# Emerging Treatments and New Vehicle Formulations for Atopic Dermatitis

**DOI:** 10.3390/pharmaceutics16111425

**Published:** 2024-11-07

**Authors:** Sibel Ali, Ana Ion, Olguța Anca Orzan, Beatrice Bălăceanu-Gurău

**Affiliations:** 1Faculty of Medicine, “Carol Davila” University of Medicine and Pharmacy, 020021 Bucharest, Romania; sibel.ali@rez.umfcd.ro (S.A.); ana.ion@rez.umfcd.ro (A.I.); beatrice.balaceanu@drd.umfcd.ro (B.B.-G.); 2Department of Dermatology, Elias University Emergency Hospital, 011461 Bucharest, Romania

**Keywords:** atopic dermatitis, novel therapies, nanotechnology, biopolymers

## Abstract

Atopic dermatitis is one of the most common inflammatory skin diseases, with an increasing incidence among both children and adults. The recurrent nature, often with the persistence of symptoms, and the polymorphism of the response to current therapies have led to increased research in the therapeutic area dedicated to this condition. The understanding of pathophysiological pathways has contributed to the development of innovative therapies, including biological therapies, JAK inhibitors, but also emerging technologies like nanotechnology-based drug delivery systems. These innovations promise enhanced efficacy, reduced side effects, and improved patient outcomes. The ongoing exploration of novel vehicles, formulations, and natural biopolymers, along with cutting-edge therapeutic agents like tapinarof and mesenchymal stem cells, highlights the potential for an even more precise and personalized management of AD in the future. Despite these advances, challenges persist, particularly in ensuring the long-term safety, accessibility, and broader application of these therapies, necessitating continued research and development.

## 1. Introduction

Atopic dermatitis (AD) is a chronic inflammatory skin disease that mainly affects children, with a prevalence of 10 to 20% [1]. More than half of cases begin during the first year of life. In adults, the condition affects approximately 2 to 10% of the population, either starting in childhood and having a chronic course, or with onset in adulthood [1,2,3,4]. The overall incidence of the disease has continued to rise, with rates currently estimated to be two to three times higher than in the 1970s [4]. This increase has significantly impacted the costs associated with medical treatment and resulted in additional expenses for patients and their families [1].

AD is a complex disease with pathophysiological mechanisms that are still not fully understood. Currently, it is recognized as a systemic, multifactorial inflammatory disease that involves both genetic predisposition and environmental factors [4]. The intricate interaction between these factors leads to a wide range of clinical presentations, varying in terms of both the type of skin lesions (including papules, vesicles, edema, crusting, and scaling) and the affected areas [2,3,4]. The clinical patterns of eczema also vary depending on the affected age group. In children, the most common areas involved are the face and the extensor surfaces of the limbs. As patients age, eczema lesions typically become more localized, often appearing in the flexural areas, particularly in the antecubital and popliteal fossae. In adults, atypical presentations are more common, with eczema frequently affecting the hands, feet, and face, especially the periocular region [2,3,4].

A central role in the pathophysiology of AD is held by immunological abnormalities, which appear to perpetuate the disease in its chronic phase and should be targeted to ensure patient improvement [5]. In AD, a type 2 immune response is engaged, which is characterized by the release of the cytokines interleukin-4 (IL-4) and interleukin-13 (IL-13) from T-helper 2 cells. These cytokines have a dual role: they both promote the inflammatory process and influence the expression of key structural proteins in the skin, ultimately affecting the skin’s barrier function [3,4,5]. Recent studies have shown that interleukin-25 (IL-25) can increase the production of IL-4 and IL-13 by innate lymphoid cells (ILC2s) and Th2 cells. IL-25 is considered a crucial “alarmin” that drives type 2 immune responses, which can worsen damage to the epithelial barrier [6]. Interleukin-31 (IL-31) has emerged as a significant mediator of pruritus by activating sensory neurons. The involvement of other cytokines, such as interleukin-21 (IL-21), which correlates with disease severity, and the transient receptor potential vanilloid channel TRPV3 [7,8,9], which mediates itch responses in lesional skin [3], further underscores the intricate network of immune signaling in AD.

Genetic factors contribute to the complexity of the pathophysiology of AD, as certain polymorphisms are associated with changes in skin barrier function and immune responses. For instance, mutations in the filaggrin gene have been linked to a higher risk of developing AD [7,8]. However, while genetic factors are significant, the global rise in AD prevalence underscores the importance of environmental influences in the disease’s development. Environmental factors that can affect the progression and severity of AD include lower average temperatures, indoor heating, reduced humidity, and a lower UV index [10,11].

An additional aspect of the complex pathophysiological mechanisms of AD is abnormalities in the skin microbiome. Specifically, an increase in *Staphylococcus aureus* has been associated with the severity of the disease. This bacterium can trigger pro-inflammatory responses and compromise the skin barrier, creating a cycle that perpetuates inflammation and skin damage.

The treatment options for atopic dermatitis (AD) include topical agents, phototherapy, and systemic medications. 

## 2. Treatment of AD

### 2.1. Topical Treatments

Prescription topical agents play a crucial role in the management of AD. These agents include moisturizers, topical corticosteroids (TCSs), topical calcineurin inhibitors (TCIs), phosphodiesterase 4 (PDE-4) inhibitors, Janus kinase (JAK) inhibitors, and novel investigational therapies (Table 1). These agents, along with their mechanisms of action, are illustrated in Figure 1 [12,13].

Topical anti-inflammatory treatments, including topical corticosteroids (TCSs) and topical calcineurin inhibitors (TCIs), play a crucial role in the management of mild-to-moderate AD. TCSs are available in a range of formulations, including ointments, creams, lotions, gels, foams, oils, solutions, and shampoos [14]. However, these traditional formulations may cause both topical and systemic side effects, which can restrict their long-term use. Common adverse effects include skin atrophy, stretch marks, rosacea, visible blood vessels, purpura, and the suppression of adrenal gland function (Table 1) [15,16].

Additionally, TCIs may cause a high incidence of local adverse symptoms, such as stinging and burning sensations upon application. They also have limited potency compared to stronger TCSs. Moreover, these medications carry a boxed warning from the U.S. Food and Drug Administration (FDA) regarding the potential risk of malignancy, despite long-term studies showing evidence to the contrary (Table 1) [15,16,17].

The profile of adverse reactions linked to TCSs and TCIs has prompted research into alternative therapies, particularly focusing on the development of topical JAK inhibitors.

The foundation of this research was based on observations on the use of oral Janus kinase (JAK) inhibitors in other autoimmune and autoinflammatory diseases, such as psoriasis, vitiligo, alopecia areata, and rheumatoid arthritis. Studies have demonstrated that JAK inhibitors can enhance skin barrier function, reduce itching, promote the elongation of cutaneous nerves, and inhibit the differentiation of Th2 cells in response to IL-4 and IL-13 [18,19]. As a result, the use of oral JAK inhibitors has also been approved for the treatment of AD.

In 2021, the FDA approved the use of topical ruxolitinib 1.5% cream, an anti-JAK 1/2 treatment, for the short-term and non-continuous management of mild-to-moderate atopic dermatitis in patients aged 12 and older. The area of application should not exceed 20% of the body surface area, and a maximum of 60 g may be applied weekly [19,20]. These guidelines are designed to minimize systemic absorption, as the product carries black box warnings related to serious infections, mortality, malignancies (such as lymphoma), major adverse cardiovascular events, and thrombosis [19,20].

Delgocitinib 0.5% ointment is another topical pan-JAK inhibitor that targets JAK1, JAK2, JAK3, and Tyk2. It received approval in Japan for the treatment of AD in adults in January 2020 and in children in March 2021. The clinical results associated with delgocitinib were shown to be comparable to those of tacrolimus 0.1% ointment [21].

Crisaborole 2% ointment is a low-molecular-weight benzoxaborole topical drug that inhibits phosphodiesterase 4 (PDE-4). It has been approved by the FDA for treating mild-to-moderate AD in patients 3 months of age and older [22].

To compare the efficacy and safety profile of these topical therapies for eczema, in 2024, Lax et al. conducted a meta-analysis in 2024. Their findings showed that potent TCSs, JAK-inhibitors (ruxolitinib 1.5%), and TCIs (tacrolimus 0.1%) consistently reduced the signs and symptoms of eczema. However, crisaborole 2% was found to be among the least effective treatments. The study also noted that side effects like burning and stinging were more likely with TCIs and crisaborole, and less likely with TCSs [22].

### 2.2. Phototherapy

In moderate-to-severe forms of AD that are insufficiently controlled with topical treatment, phototherapy and/or systemic therapies are recommended [23]. These are illustrated in Figure 2, along with their mechanisms of action.

Phototherapy is a treatment method indicated for various conditions, including for patients over 6 years old with moderate-to-severe forms of AD [23].

Narrowband ultraviolet B (UVB) radiation is the most commonly used type of phototherapy for treating inflammatory skin diseases. This treatment is typically administered two to three times per week over a duration of 10 to 14 weeks. Potential side effects of phototherapy can include reactions similar to sunburn, such as erythema, blistering, and the possible reactivation of herpes simplex infection [24].

### 2.3. Biologic Drugs

The FDA has approved two biologic treatments for AD: dupilumab, which is an IL-4 and anti-IL-13 inhibitor, and tralokinumab, which is an anti-IL-13 monoclonal antibody.

Dupilumab is the first biologic agent approved for the treatment of moderate-to-severe forms of AD in patients over 6 months old [25]. Its target is the α-subunit of the IL-4 receptor, which is part of both the IL-4 and IL-13 receptor complexes [23]. Dupilumab has demonstrated particular efficacy in patients with Th2-related conditions, such as asthma, allergic rhino-conjunctivitis with nasal polyps, and eosinophilic esophagitis [23,24]. Its use is sometimes limited by the occurrence of side effects, particularly conjunctivitis [23] (Table 2).

In 2021, the European Medicines Agency (EMA) approved the second biologic drug for treating moderate-to-severe AD in patients aged 16 and above, tralokinumab. This is an IgG4 monoclonal antibody targeting IL-13 that results in significant improvement in the signs and symptoms of AD, as well as in quality of life (QOL) [26]. The safety profile of tralokinumab is similar to that of dupilumab, but it is associated with fewer ocular adverse reactions [26]. There is currently no direct comparative data on the efficacy of tralokinumab compared to other systemic therapies. However, network meta-analyses indicated that, after 16 weeks of treatment, tralokinumab’s efficacy was lower than that of dupilumab. This evaluation was based on the Eczema Area and Severity Index (EASI) score [24,25].

In November 2023, lebrikizumab received approval from the EMA for the treatment of AD in patients aged 12 and older who are candidates for systemic therapy. This agent is a humanized IgG4 monoclonal antibody that binds soluble IL-13 [27]. Its efficacy was demonstrated in a phase 3 clinical trial in adolescents and adults with moderate-to-severe forms of AD at week 16, both as a standalone treatment and in combination with TCS [28].

Although tralokinumab and lebrikizumab are both anti-IL-13 agents, there are subtle differences in their mechanisms of action. Tralokinumab blocks the binding of IL-13 to both the IL-13Rα1 and IL-13Rα2 decoy receptors. This action is believed to play a role in the endogenous regulation of IL-13. On the other hand, lebrikizumab selectively prevents the formation of the IL-13Rα1/IL-4Rα receptor signaling complex, while still allowing IL-13 to bind to the IL-13Rα2 receptor.

Another anti-IL-13 agent that is currently under investigation is cendakimab. According to the results of a phase 2 randomized clinical trial, cendakimab demonstrated efficacy by reducing the EASI score at week 16 of treatment. The study also showed a safety profile for cendakimab similar to that of other biological therapies [29].

### 2.4. JAK-Inhibitors

JAK inhibitors are increasingly being recognized as important treatments for various systemic diseases, including rheumatoid arthritis, psoriatic arthritis, alopecia areata, inflammatory bowel disease, and more recently, atopic dermatitis [30,31,32].

The JAK kinase family comprises four receptor-associated kinases: JAK1, JAK2, JAK3, and TYK2 (tyrosine kinase 2). The STAT family includes seven proteins: STAT1, STAT2, STAT3, STAT4, STAT5A, STAT5B, and STAT6. The JAK-STAT signaling cascade is initiated when extracellular cytokines bind to specific receptors on the cell membrane. Consequently, these enzymes phosphorylate proteins involved in the signaling cascade (STATs), which then translocate into the cell nucleus to participate in gene transcription [31,32].

In the pathophysiology of AD, the Th2 immune response is overexpressed, particularly in the acute phase. The interleukins involved in the Th2 immune response include IL-4, IL-5, IL-13, and IL-31. IL-4 regulates Th2 target genes by signaling through JAK1, JAK3, and STAT6, while IL-13 signals through JAK1, JAK2, TYK2, and STAT6 to activate IL-13-responsive genes. IL-31 signals through JAK1, JAK2, STAT1, STAT3, and STAT 1 [31].

JAK inhibitors target one or more JAK enzymes, thereby inhibiting the intracellular signaling cascade and downstream inflammatory processes. These molecules are therefore of significant interest in diseases where the immune system is overactive, such as rheumatoid arthritis, psoriatic arthritis, ulcerative colitis, and other autoimmune disorders [31,32].

Upadacitinib and abrocitinib are oral JAK inhibitors that selectively target JAK1 and are approved for patients with moderate-to-severe AD who have not responded to, or have contraindications for, other systemic treatments [25]. Baricitinib is a JAK1 and JAK2 inhibitor approved in Europe for moderate-to-severe AD, particularly in patients with other inflammatory comorbidities such as rheumatoid arthritis or alopecia areata. Network meta-analyses indicate that baricitinib is less effective than upadacitinib and abrocitinib [23,25].

The response rates for achieving at least a 75% improvement in clinical scores with monotherapy are approximately 35% for baricitinib, 60% for abrocitinib, and 75% for upadacitinib (at the highest dose for each) [32,33].

Pharmacologically, JAK inhibitors are administered orally, have a short half-life, and exhibit a rapid onset of action [32]. Reported adverse effects include an increased frequency of upper-respiratory-tract infections, herpes simplex, the reactivation of varicella-zoster virus, elevated creatine phosphokinase levels, and nausea [34]. However, no significant increase has been observed in severe infections, non-melanoma skin cancers, other malignancies, severe cardiovascular adverse effects, venous thromboembolism, or nasopharyngitis (Table 2) [34].

To minimize the risk of serious adverse effects, JAK inhibitors are recommended only when no alternative therapeutic options are available in patients over 65 years of age or in those with a high risk of cardiovascular complications or malignancies. Prior to initiating these therapies, latent infections such as tuberculosis or viral hepatitis, organ dysfunctions like hepatic or renal impairment, and pregnancy should be ruled out [25,34]. JAK inhibitors are approved or under investigation for the treatment of multiple conditions, including rheumatoid arthritis, psoriatic arthritis, alopecia areata, inflammatory bowel disease, and more recently, AD [30,31,32].

JAK inhibitors work by blocking the JAK-STAT intracellular signal transduction pathway. The JAK family, including JAK1, JAK2, JAK3, and TYK2, are cytoplasmic tyrosine kinases that bind to various cytokine receptors to form signaling complexes [31]. These complexes regulate inflammation by activating transcription factors known as STATs (signal transducer and activator of transcription), which in turn influence the expression of genes related to inflammation. IL-4 and IL-13 predominantly engage in sustained activation via JAK1/2-STAT6, while TSLP and IFN-γ are implicated in persistent activation with JAK1/2-STAT5 [31,32]. These inflammatory factors enhance Th2 immune activity, thereby triggering Th2-biased inflammatory responses [32]. Inhibiting JAK activity can be more effective than targeting individual cytokine pathways.

Upadacitinib and abrocitinib are selective oral JAK inhibitors that primarily target JAK-1. They are approved for patients with moderate-to-severe AD who have not responded to other systemic treatments (such as immunosuppressants, corticosteroids, antimetabolites, or injectable biologics), or when these treatments are not suitable [25].

Baricitinib selectively inhibits JAK-1 and JAK-2, is approved in Europe for treating moderate-to-severe AD, and is available in the U.S. for other immune-mediated conditions, though it is not FDA-approved for AD. It is especially recommended for patients with AD suffering from inflammatory comorbidities, such as rheumatoid arthritis or alopecia areata. Although no direct clinical trials have been conducted, network meta-analyses indicate that baricitinib is less effective than upadacitinib and abrocitinib [23,25].

JAK inhibitors are administered orally; they have a short half-life and a rapid onset of action [32]. The response rates for achieving at least a 75% improvement in clinical scores when used as monotherapy are approximately 35% for baricitinib, 60% for abrocitinib, and 75% for upadacitinib (at the highest dose in each case) [32,33].

Side effects reported in studies on the use of JAK inhibitors to treat AD include an increased frequency of upper-respiratory-tract infections, herpes simplex, and varicella zoster reactivation. Patients at risk should, therefore, be vaccinated against herpes zoster. Transient nausea has been described more frequently with abrocitinib, and transient acneiform skin manifestations have been described with upadacitinib [34] (Table 2).

A recent meta-analysis on the safety of JAK inhibitors showed that the risk of herpes zoster, headache, acne, elevated blood creatinine phosphokinase, and nausea was significantly increased, but the risk of serious infection, non-melanoma skin cancer (NMSC), malignancies other than NMSC, major adverse cardiovascular events, venous thromboembolism, and nasopharyngitis was not increased [34]. Nevertheless, to minimize the risk of serious adverse events, JAK inhibitors should not be used in persons over age 65, persons at increased risk of serious cardiovascular problems or cancer, or for current or past smokers, unless there is no good alternative treatment. Before treatment with JAK inhibitors is started, latent infections such as tuberculosis and hepatitis, marked renal or hepatic dysfunction, and pregnancy must be ruled out [25]. Women of childbearing age must use effective contraception while being treated with JAK inhibitors [25,34].

### 2.5. Antimetabolites and Immunosuppressants

The most commonly recommended conventional systemic therapies for patients with moderate-to-severe forms of AD are cyclosporine and methotrexate. There are also other agents which are infrequently used, such as azathioprine and mycophenolate mofetil. These medications require initial and periodic monitoring of laboratory tests due to potential adverse effects. Each drug is associated with specific end-organ toxicities: cyclosporine can lead to renal impairment and hypertension, methotrexate can cause liver test abnormalities, and both azathioprine and mycophenolate mofetil may cause cytopenia. It is also important to note that cyclosporine is not recommended for long-term use because the risk of renal damage increases with cumulative dosage [25]. During these treatments, patients have an increased risk of infections due to the immunosuppressive effect, some of which can be potentially severe (Table 2).

While systemic corticosteroids are often prescribed in clinical practice for patients with moderate-to-severe AD, their use should generally be avoided, particularly in children. This decision is justified by both the lack of long-term remission and the profile of adverse reactions (Table 2) [33]. Additionally, the efficacy of systemic corticosteroids compared to cyclosporine is significantly lower [35]. In certain limited situations, clinicians may consider short courses of systemic corticosteroids, such as when no other treatment options are available, or as a temporary measure before initiating long-term therapies [25].

## 3. Novel Vehicles and Formulations

Since the therapies currently available for AD are associated with various types of adverse reactions, ranging from local ones, such as erythema, atrophy, and tachyphylaxis, to systemic ones—immunosuppression, increased risk of infections, conjunctivitis, etc.—the development of new and innovative therapies is mandatory. These new drugs should address the limitations of traditional formulations and enhance therapeutic effectiveness. New topical vehicles and formulations, such as nanotechnologies, natural biopolymers like chitosan, polyaphron dispersion (PAD) technology, and many others have attracted significant attention in this field (Figure 3).

### 3.1. Topical JAK Inhibitors

Gusacitinib (GUSA) is a selective inhibitor of JAK and SYK (spleen tyrosine kinase). A preliminary phase 1b randomized clinical trial indicated that GUSA is safe and well tolerated. The most frequently reported adverse reactions were headache, nausea, and diarrhea, with the placebo group experiencing them more often [36].

Brepocitinib (BREPO) is a JAK1/Tyk2 selective inhibitor with promising results in treating patients with moderate-to-severe AD, available in both topical and oral forms. A randomized phase 2b clinical trial involving patients aged 12 to 75 with moderate-to-severe AD found that applying BREPO 1% cream twice daily resulted in a 75% reduction in the EASI score, compared to a 47.6% reduction in the placebo group. Similar results were observed with once-daily applications of BREPO formulations at 3%, 1%, and 0.3%. The trial also met its secondary objectives, including achieving EASI-75, Investigator’s Global Assessment (IGA) scores of 0 or 1, and a reduction in itch as measured by the Numeric Rating Scale for Itch (NRS4). Regarding safety, no significant adverse events were noted, and the incidence of side effects was lower in the BREPO group compared to the placebo group [37].

Currently, additional topical JAK inhibitors are being tested in phase 2 studies: ATI-1777, jaktinib, SHR0302, and ATI-502 [36].

### 3.2. Recent Nanotechnologies

Nanotechnology is a branch of engineering that uses nanoscale particles (100 nm and smaller). The field is constantly evolving and has various applications in the medical domain, including the treatment of patients suffering from AD. These innovative nanoparticle-based systems have demonstrated potential for delivering drugs through the skin by improving drug penetration across the stratum corneum. This layer of the skin is a natural barrier against particle penetration, being typically impermeable to many substances. Several examples of nanotechnology applications for the treatment of AD highlight their potential to enhance drug delivery and therapeutic efficacy.

Polymeric nanoparticles, including those made from chitosan, poly-lactic-co-glycolic acid (PLGA), and poly-caprolactone, are promising for delivering drugs for AD [38]. They enhance drug solubility, stability, and skin permeability.

Lipid-based nanosystems (LBNs) are formed through a lipid phase and surfactants. They have demonstrated that they improve the delivery of various active principles to specific skin layers, with stated localization in the upper layers of the skin [39]. There are different kinds of LBN: liposomes, ethosomes, transferosomes, solid lipid nanoparticles, nanostructured lipid carriers, cubosomes, and monoolein aqueous dispersions [39].

A recent small-scale study showed that H.ECMTM liposome emollient, which contains a soluble proteoglycan, hydrolyzed collagen, and hyaluronic acid (HA), demonstrated important results in improving itching severity and transepidermal water loss in patients with mild atopic dermatitis and dry skin [40]. The emollient, which contains 0.01% H.ECMTM liposomes, consists of a novel formula comprising a soluble proteoglycan derived from salmon nasal cartilage combined with non-denatured N-terminus emulsified HA and hydrolyzed collagen enclosed in nano-liposomes. Twenty-five participants used the moisturizer on their affected skin at least twice daily for four weeks, and the results indicated noticeable improvements in clinical symptoms and biophysical parameters related to skin health [40].

In another study, Slavkova M et al. successfully integrated nanoparticles containing budesonide into two different hydrogels (methylcellulose and Pluronic F127), which allowed for easy application and effective hydration in topical formulations [41]. Both hydrogels demonstrated their suitability for dermal use based on their ability to spread, penetrate, maintain pH balance, and provide occlusion. The developed nanocarriers showed no signs of irritation when tested on human keratinocyte cell lines in vitro, indicating their potential safety for topical application [41].

### 3.3. Natural Biopolymers

Natural biopolymers are recognized for their unique characteristics that make them suitable for drug delivery. These attributes include biocompatibility, biodegradability, low cost, and bioactivity. These polymers can be incorporated into nanocarriers or semi-solid hydrogel matrices, resulting in hybrid systems. Examples include chitosan nanoparticles (NPs), starch NPs, and liposomes used in creams or gels to enhance the delivery of corticosteroids, hydroxytyrosol, or phytoceramides for managing AD [38].

Chitosan, one of the most well-known biopolymers, is obtained through the deacetylation of chitin, a natural polymer found in the exoskeletons of crustaceans and the cell walls of fungi [42,43,44]. This process removes acetyl groups, resulting in a more soluble polymer with distinct properties compared to its precursor. Chitosan has several advantages, including biocompatibility, biodegradability, antimicrobial properties, antioxidative properties, and inflammatory response modulation, making it beneficial for treating conditions like atopic dermatitis [44]. Furthermore, its versatile formulation allows it to be used in various delivery systems, including hydrogels, films, nanoparticles, and textiles, facilitating diverse applications in drug delivery and tissue engineering [42,43,45].

Chitosan could prove useful in the preparation of nanoparticles designed to deliver various active substances that have demonstrated efficacy in treating patients with AD, such as hydrocortisone, hydroxytyrosol combination, betamethasone, and tacrolimus [42].

While chitosan offers numerous advantages for drug delivery and therapeutic applications, challenges related to mechanical properties, production complexity, and clinical validation remain areas for further investigation.

## 4. Emerging Advanced Technologies and Molecules

Among the advanced technologies and molecules currently under investigation are curcumin, peptide-based therapies, aryl hydrocarbon receptor agonists, and TRPV-1 antagonists (Figure 4).

Curcumin has powerful anti-inflammatory properties that have caught the attention of researchers, establishing it as a natural compound with potential therapeutic benefits for different inflammatory skin conditions, including AD [46,47,48]. In a recent study, topical formulations incorporating curcumin within a self-nanoemulsifying drug delivery system (SNEDDS) effectively reduced levels of TNF-α and IL-1β, as demonstrated in adult human epidermal cell experiments, making these preparations promising for treating inflammatory skin conditions, including AD [46].

Lim et al. conducted a study on a new peptide-based therapy that utilizes an intra-dermal delivery technology (IDDT) platform [49]. This platform evaluates human-derived peptides with cell-penetrating peptide (CPP) properties. Among the studied peptides, the RMSP1 peptide exhibited strong anti-inflammatory effects by targeting the NF-κB and STAT3 signaling pathways. Additionally, formulations using liposomes and micelles demonstrated improved anti-inflammatory effects in laboratory studies and enhanced therapeutic results in live subjects [49]. These findings suggest that nano-formulated RMSP1 shows potential as a new treatment option for atopic dermatitis.

Aryl hydrocarbon receptors (AhRs) are expressed in skin cells, including keratinocytes and dendritic cells. They can be activated by a wide range of external and internal molecules and play a role in mediating epidermal differentiation [50]. By activating AhR signaling pathways, tapinarof modulates gene expression, leading to the downregulation of type 2 inflammation (IL-4, -13), the normalization of the skin barrier, and reductions in oxidative stress. In May 2022, the FDA approved the use of tapinarof cream 1% as an AhR agonist for treating plaque psoriasis in adults, and in April 2024, tapinarof 1% received the Supplemental New Drug Application (sNDA) for the topical treatment of AD in adults and children 2 years of age and older, following positive findings in the phase 3 clinical trial [50,51].

TRPV-1 is present in keratinocytes, dendritic cells, mast cells, and sensory neurons, and it is upregulated in the affected skin of individuals with AD. While TRPV-1 influences inflammation, it has been demonstrated to facilitate both histamine-dependent and -independent itch by releasing central neuropeptides like substance P (SP) and calcitonin gene-related peptide (CGRP) upon activation. In a randomized, vehicle-controlled phase 3 study involving patients aged 12 and older with mild-to-moderate AD, the selective TRPV-1 antagonist asivatrep exhibited efficacy in improving eczematous lesions and alleviating itch [52].

Mesenchymal stem cells (MSCs), which have demonstrated effectiveness in treating various conditions in preclinical studies, are also currently being investigated in clinical trials for patients with moderate-to-severe forms of AD. The product under study in patients aged 19 years and above with moderate-to-severe AD is hUCB-MSsc/FURESTEM-AD^®^ (human umbilical cord blood-derived mesenchymal stem cells). In mice, the application of hUCB-MSsc resulted in a reduction in elevated serum IgE levels and mast cell degranulation. Phase 1/2a trials showed a 55% improvement in the EASI score and a 58% reduction in pruritus at week 12. Improvement in skin lesions and itch was observed after two weeks. No serious adverse events were reported, just local skin reactions following weekly subcutaneous injections, skin infections, and a gastrointestinal adverse event [53].

Polyaphron dispersion (PAD) technology primarily involves oil-in-water dispersions containing oil droplets ranging from 1 to 50 μm [54]. In PAD formulations, the droplets consist of an inner core made of a nonpolar solvent, usually a pharmaceutically acceptable oil. During the manufacturing process, this core is encapsulated by an outer shell that features a multi-layer structure composed of surfactants, oil, and water. The key benefits of PAD technology compared to traditional topical formulations include enhanced drug penetration, local tolerability, convenience of use, and stability of the active ingredients. The multi-molecular shell structure of PAD formulations protects potentially unstable active molecules from hydrolytic degradation, ensuring that the therapeutic agents remain effective over time [54]. The challenges and limitations of using PAD technology may include manufacturing difficulties to ensure stable dispersions. Additionally, these systems may not be compatible with all active ingredients, which could restrict the use of this technology. Environmental factor variations could also alter the stability of active ingredients, posing further challenges in the production process. Finally, the high costs associated with producing these new technologies may impact their potential for widespread use [54].

The combination of synthetic pseudo-ceramide (SLE66) and eucalyptus leaf extract (ELE) has been found to enhance ceramide synthesis in the skin [55]. When used in a skin moisturizer, this combination has been shown to improve the barrier functions and water-holding capacity of skin affected by AD, leading to an improvement in skin symptoms. ELE has been found to increase ceramide levels in human keratinocytes and the stratum corneum when applied topically. The skin moisturizer containing SLE66 and ELE, known as P-Cer moisturizer, is effective for mild-to-moderate AD, with more than 60% of patients experiencing significant improvement in their skin symptoms. The treatment has also been found to reduce sensitivity to mite antigens, decrease erythema and itchiness, and improve skin dryness [55].

Protein degradation targeting chimeras (PROTACs) are a new category of small bifunctional molecules that can induce the ubiquitin–proteasome system (UPS)-mediated degradation of target proteins [56]. The inhibitory activity of JAPT, a PROTAC-based small-molecule JAK1/JAK2 degrader on the JAK-STAT pathway, was investigated using both in vitro and in vivo inflammatory models. JAPT, as a therapeutic agent against AD, demonstrated successful initiation of the JAK ubiquitination–degradation program, leading to the inhibition of the JAK-STAT pathway. Notably, when administered topically, JAPT showed greater potency compared to JAK inhibitors, consistent with the expected target effects [56].

## 5. Conclusions

The treatment landscape for AD has significantly evolved, with advances in both topical and systemic therapies aimed at addressing the complex pathophysiology of the disease. Traditional treatments such as corticosteroids and calcineurin inhibitors remain essential, but are now complemented by newer options like JAK inhibitors, biologics targeting key cytokines (IL-4, IL-13), and emerging technologies like nanotechnology-based drug delivery systems. These innovations promise enhanced efficacy, reduced side effects, and improved patient outcomes. The ongoing exploration of novel vehicles, formulations, and natural biopolymers, along with cutting-edge therapeutic agents like tapinarof and mesenchymal stem cells, highlights the potential for an even more precise and personalized management of AD in the future. Despite these advances, challenges persist, particularly in ensuring the long-term safety, accessibility, and broader application of these therapies, necessitating continued research and development.

## Figures and Tables

**Figure 1 pharmaceutics-16-01425-f001:**
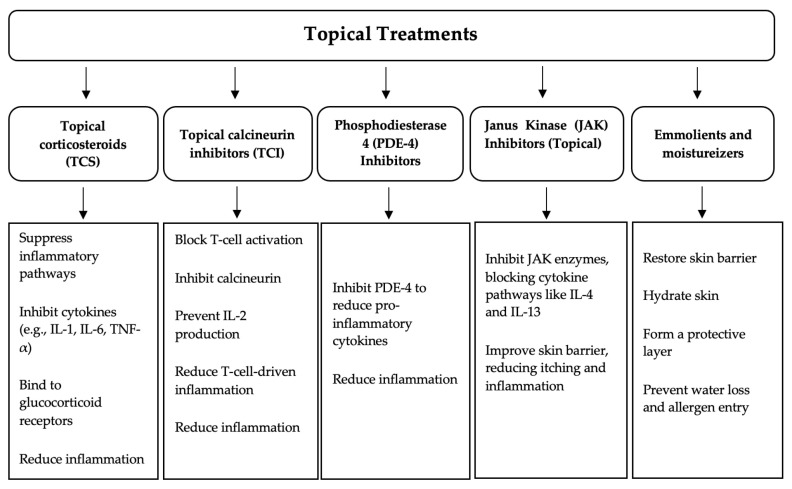
Topical treatments for AD.

**Figure 2 pharmaceutics-16-01425-f002:**
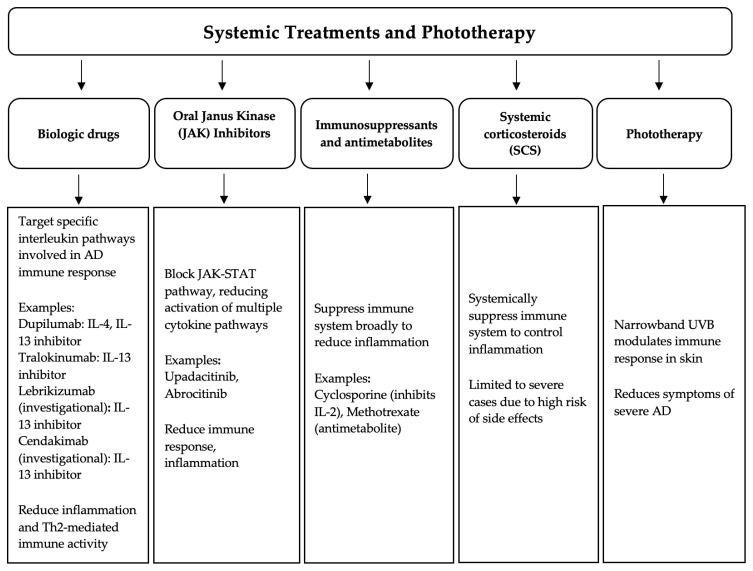
Mechanism of action of systemic agents and phototherapy used for the treatment of patients with AD.

**Figure 3 pharmaceutics-16-01425-f003:**
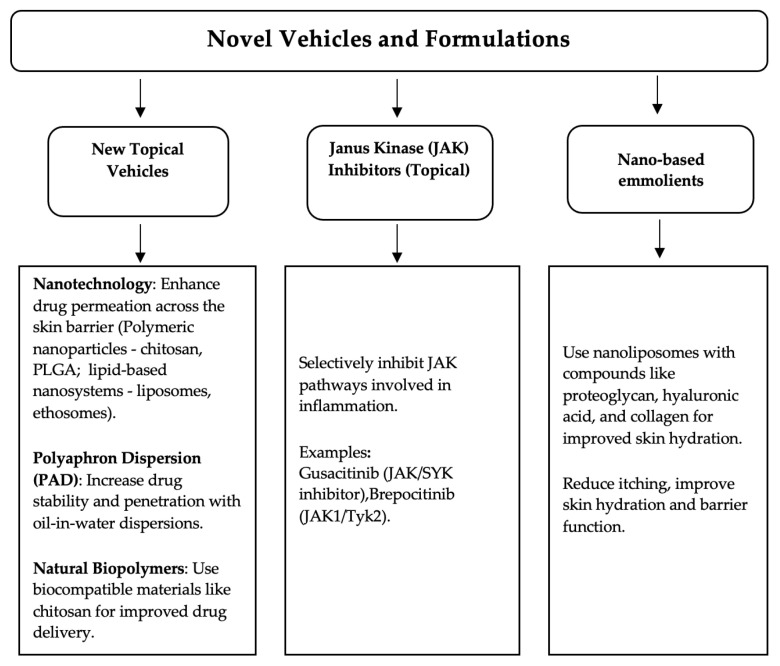
Novel vehicles and formulations indicated for AD.

**Figure 4 pharmaceutics-16-01425-f004:**
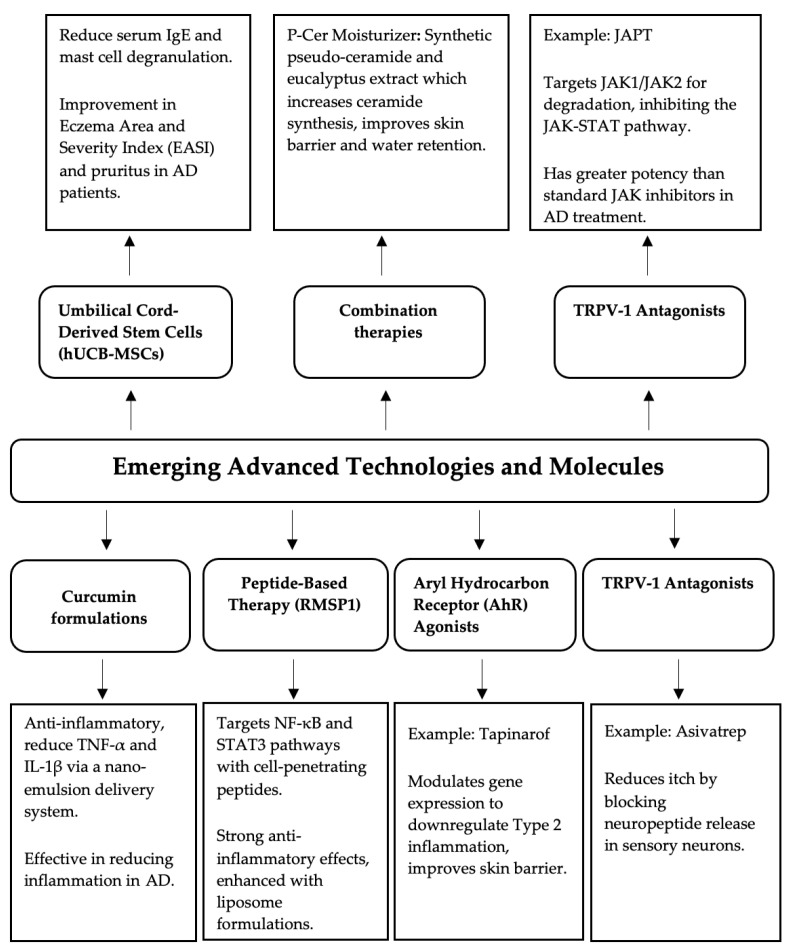
Emerging advanced technologies and molecules used in the treatment of AD.

**Table 1 pharmaceutics-16-01425-t001:** Topical treatments for AD and their common adverse reactions.

Topical Treatment	Type	Common Adverse Reactions
Topical corticosteroids (TCSs)	Anti-inflammatory agent	Skin atrophy, striae, rosacea, telangiectasia, purpura, adrenal suppression (with prolonged use)
Topical calcineurin inhibitors (TCIs), e.g., tacrolimus, pimecrolimus	Anti-inflammatory agent	Burning and stinging at the application site, potential malignancy risk (FDA boxed warning)
Crisaborole (PDE-4 inhibitor)	Anti-inflammatory agent	Burning and stinging at the application site
Topical ruxolitinib (JAK 1/2 inhibitor)	JAK inhibitor	Application site reactions (e.g., burning, itching, redness), acne, nasopharyngitis, headache

**Table 2 pharmaceutics-16-01425-t002:** Systemic treatments and their common adverse reactions.

Systemic Treatment	Type	Common Adverse Reactions
Dupilumab	Biologic (anti-IL-4, IL-13)	Conjunctivitis, injection site reactions, transient eosinophilia, recurrence of IL-17 diseases (e.g., psoriasis)
Tralokinumab	Biologic (anti-IL-13)	Fewer ocular complications than dupilumab, injection site reactions
Lebrikizumab	Biologic (anti-IL-13)	Injection site reactions, headache, upper respiratory infections
Upadacitinib	JAK-1 inhibitor	Acneiform skin eruptions, increased risk of herpes zoster, upper respiratory infections, headache, nausea
Abrocitinib	JAK-1 inhibitor	Nausea, upper-respiratory-tract infections, herpes zoster reactivation
Baricitinib	JAK-1/2 inhibitor	Increased risk of infections, including herpes zoster, elevated creatinine kinase, headache
Cyclosporine	Immunosuppressant	Nephrotoxicity, hypertension, increased risk of infections, gingival hyperplasia, hypertrichosis
Methotrexate	Antimetabolite	Liver toxicity, bone marrow suppression, gastrointestinal disturbances, lung toxicity
Azathioprine	Immunosuppressant	Bone marrow suppression, increased risk of infections, gastrointestinal disturbances
Mycophenolate mofetil	Immunosuppressant	Bone marrow suppression, gastrointestinal issues, increased risk of infections
Systemic corticosteroids	Corticosteroid	Weight gain, hypertension, hyperglycemia, osteoporosis, adrenal suppression, increased risk of infections

## Data Availability

Not applicable.

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
