# Peer review of "Emerging Treatments and New Vehicle Formulations for Atopic Dermatitis"

_pharmaceutics, 2024, doi:10.3390/pharmaceutics16111425_

Round 1
Reviewer 1 Report
Comments and Suggestions for Authors
Dear Author,
The manuscript provides a comprehensive overview of the evolving treatment landscape for atopic dermatitis (AD), highlighting the advancements in both topical and systemic therapies. The article is well written and the content provided were found to be satisfactory. However, there are few contents which might be included like
1. In introduction - the prevalence and causes of atopic dermatitis found to be incomplete. Ensure each section, particularly in the Introduction and Innovative Therapies, is fully developed with complete sentences and detailed explanations
2. Insights on topical JAK inhibitors and technologies like Polyaphron Dispersion are incomplete. Provide more detailed discussions on the mechanisms of biological therapies and JAK inhibitors. Discuss the implications of nanotechnology-based delivery systems in greater detail, including potential benefits and limitations.
3. The manuscript outlines challenges but lacks detailed strategy or solutions
4. Ensure all references are complete and follow the same citation style for consistency. Add more recent and relevant references to support the statements made in the manuscript.
5. The manuscript does not provide a detailed comparative analysis of the efficacy, safety, and cost-effectiveness of the various treatment options discussed, which could be useful for clinicians and patients in decision-making.
6. The manuscript does not delve into the potential barriers to the widespread adoption of these innovative therapies, such as regulatory hurdles, manufacturing complexities, or healthcare system limitations, which could impact their real-world implementation.
The author may try to include the comments stated based on the available literatures and resources.
Reviewer 2 Report
Comments and Suggestions for Authors
1. Authors discussed the existing therapies in detail without much concentrating in the novel treatment strategies. The recent done studies can be more elaborated with the help of tables if any. Most of the details of the existing therapies are available in most of the literatutres.
2. There can be one figures demonstrating mechanism of action of the conventional treatment and action of newer therapies.
3. Check all the biological/scientific names written in italics for e.g., Staphylococcus aureus.
4. Check the spacing “biopolymers( have unique”, typographical mistakes “for vari skin conditions” “for various skin conditions”
5. The topical agents, phototherapy, and systemic medication should come under subtopics under headings:
2. Therapeutic treatment for AD
2.1. Topical agents
2.2. Phototherapy……………and so on
